# Evaluation of Satellite-Based Rainfall Estimates in the Lower Mekong River Basin (Southeast Asia)

**Chelsea Dandridge** [1,*], **Venkat Lakshmi** [1], **John Bolten** [2] **and Raghavan Srinivasan** [3]

1   Department of Engineering Systems and Environment, 151 Engineers Way PO Box 400747,
    Olsson Hall Room 101E, University of Virginia, Charlottesville, VA 22904, USA; vlakshmi@virginia.edu
2   Hydrological Sciences Lab, NASA Goddard Space Flight Center, Greenbelt, MD 20771, USA;
    john.bolten@nasa.gov
3   Department of Ecosystem Science and Management, Texas A&M University,
    College Station, TX 77843, USA; r-srinivasan@tamu.edu
*   Correspondence: cld9mt@virginia.edu

**Abstract:** Satellite-based precipitation is an essential tool for regional water resource applications that requires frequent observations of meteorological forcing, particularly in areas that have sparse rain gauge networks. To fully realize the utility of remotely sensed precipitation products in watershed modeling and decision-making, a thorough evaluation of the accuracy of satellite-based rainfall and regional gauge network estimates is needed. In this study, Tropical Rainfall Measuring Mission (TRMM) Multi-Satellite Precipitation Analysis (TMPA) 3B42 v.7 and Climate Hazards Group InfraRed Precipitation with Station data (CHIRPS) daily rainfall estimates were compared with daily rain gauge observations from 2000 to 2014 in the Lower Mekong River Basin (LMRB) in Southeast Asia. Monthly, seasonal, and annual comparisons were performed, which included the calculations of correlation coefficient, coefficient of determination, bias, root mean square error (RMSE), and mean absolute error (MAE). Our validation test showed TMPA to correctly detect precipitation or no-precipitation 64.9% of all days and CHIRPS 66.8% of all days, compared to daily in-situ rainfall measurements. The accuracy of the satellite-based products varied greatly between the wet and dry seasons. Both TMPA and CHIRPS showed higher correlation with in-situ data during the wet season (June–September) as compared to the dry season (November–January). Additionally, both performed better on a monthly than an annual time-scale when compared to in-situ data. The satellite-based products showed wet biases during months that received higher cumulative precipitation. Based on a spatial correlation analysis, the average r-value of CHIRPS was much higher than TMPA across the basin. CHIRPS correlated better than TMPA at lower elevations and for monthly rainfall accumulation less than 500 mm. While both satellite-based products performed well, as compared to rain gauge measurements, the present research shows that CHIRPS might be better at representing precipitation over the LMRB than TMPA.

**Keywords:** remote sensing precipitation; satellite validation; Lower Mekong River Basin; water resource management

## 1. Introduction

Precipitation is one of the most important features in the global water and energy system and is vital to effective hydrology and climate research [1]. The Lower Mekong River Basin (LMRB) in Southeast Asia is particularly susceptible to precipitation-based natural disasters and is heavily dependent on proper water resource management to adequately sustain the more than 60 million inhabitants in the region whose livelihoods depend on the food and agriculture provided by the

Mekong River and its many tributaries [2]. The Mekong River is considered the tenth largest river in the world based on discharge and length [3]. In this region, rainfall seasonality causes droughts and floods that can negatively affect local resources associated with fishing and farming [4]. In the Mekong River Basin, the Northwest monsoon is responsible for the dry season and its cooler temperatures, which lasts from November to February, while the Southwest monsoon brings the wet season and warmer temperatures from June to September [5]. Precipitation in the lower basin follows an east to west gradient with the highest annual rainfall accumulation (3000 mm) occurring in the uplands of Laos and Cambodia and the least accumulation (1300 mm) occurring in northeast Thailand [4]. Being the most important river in southeast Asia, the Mekong is a significant water source and provides renewable energy and food security to people in the countries of China, Myanmar, Cambodia, Laos, Thailand, and Vietnam (Figure 1) [6,7]. Increasing development and population demand alongside changes in climate could threaten the important resources in this region if the water resources are not properly monitored and managed [5].

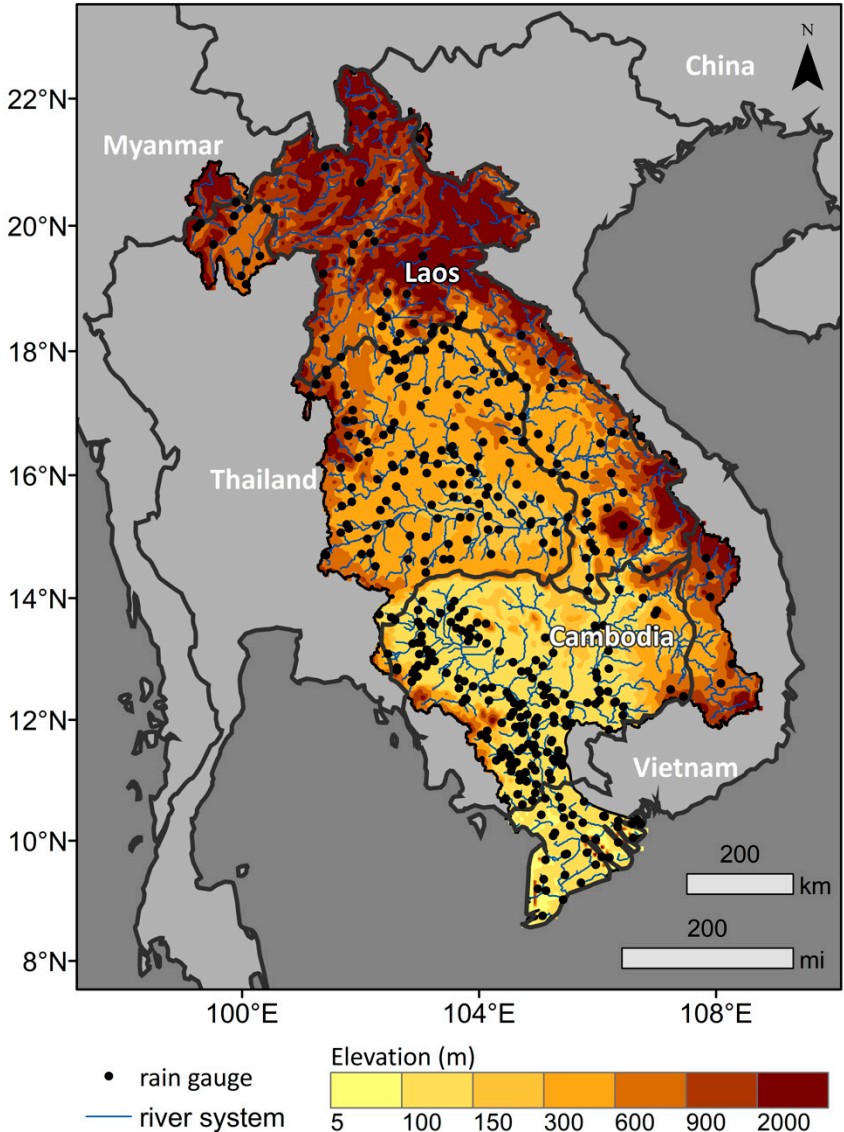

**Figure 1.** Map of the Lower Mekong River Basin in Southeast Asia and the locations of the rain gauge stations within the basin.

Across the globe, precipitation is typically estimated via three methods—ground-based rain gauges, ground radars, and satellite remote sensing [8]. Traditionally, rain gauges are used to measure rainfall due to their accuracy and reliability [9]. In this region, most of the rainfall used for decision-making purposes is measured directly by a multitude of rain gauge stations that cannot effectively reflect the spatial variation of precipitation due to its uneven distribution and limited representation as point measurements (Figure 1) [10]. Even though rain gauge stations provide the most accurate precipitation measurements, several sub-basins in the LMRB do not contain any gauges. Most sub-basins have very few or no stations and within the basins that do have stations, some stations might only record data for certain years or have time gaps in the data recording. The quality of data and techniques for data collection vary throughout the basin, which leads to a precipitation data set with significant gaps and high latency [9]. With such high limitations, it can take years to obtain suitable data for research [10]. Ground radar systems can be useful in providing the spatial distribution needed for effective precipitation estimates at the basin-scale but could also have shortcomings due to limited area coverage, high costs, and requirements of extensive equipment maintenance [11]. Therefore, in this study no ground radar data were used. Several studies indicate that high resolution satellite products are an effective alternative to ground and radar methods [12,13]. Allowing for continuous and repetitive rainfall measurements, remotely sensed satellite precipitation estimates are, thus, very useful in the LMRB due to its large geographic extent [14]. Utilizing remote sensing products and models is essential for addressing hydrological issues in the Mekong region [15,16]. To effectively utilize satellite-observed precipitation products, their accuracies should be examined over various spatial extents and time periods [14]. Knowledge of their uncertainties over varied terrain will also help us to obtain a better understanding of their applications and limitations in hydrologic models [17].

The Climate Hazards group Infrared Precipitation with station observations (CHIRPS) and NASA's Tropical Rainfall Measuring Mission (TRMM) Multi-Satellite Precipitation Analysis (TMPA) 3B42 v.7 were evaluated in this study [18,19]. It is important to note that the TRMM products applied in this study are being phased out and replaced by Global Precipitation Mission (GPM) Integrated Multi-Satellite Retrievals for GPM (IMERG) product which has improved spatial and temporal resolution, i.e., 0.1° by 0.1° and half-hourly temporal resolution [20]. Thus, the results shown here are not expected to correlate closely with GPM IMERG. However, an interesting caveat is that the TRMM combined product was used as a calibration standard for the GPM IMERG algorithm, thus, warranting this inspection of the TMPA product over the Lower Mekong River Basin [20].

Being a relatively new precipitation product, CHIRPS has been involved in limited precipitation validation studies, but was found to correlate well with in-situ measurements [21–23]. These studies are different from our work presented here regarding methodology, study period, and region. For example, an evaluation of CHIRPS was performed by Guo et al. (2017) in the LMRB, but used only 38 rain gauge stations from the Global Summary of the Day (GSOD) for validation of the satellite-based product [24]. They used the criteria of at having 30% or less missing values of the time-series record for considering individual rain gauges. This study found that CHIRPS was able to properly estimate periods of low rainfall that are associated with droughts in the region. An extensive comparison of CHIRPS was evaluated over mainland China by Bai et al. (2018) and used 2480 stations for validation from 1981 to 2014 [25]. These authors evaluated the spatio-temporal aspects of CHIRPS and found it to perform better for large rainfall amounts than arid or semi-arid regions and found a strong relationship between CHIRPS and monsoon movement. Additionally, in this study, CHIRPS was found to perform better in the warm months than winter months due to its limited capability to detect snow [25]. Similarly, several studies have found that TMPA was helpful in addressing a multitude of hydrological problems, such as predicting and monitoring precipitation [26,27]. A similar study by Wang et al. (2017) compared TMPA and GPM precipitation products over the entire Mekong basin, but only used data from 53 rain gauge stations over a 2-year period from 2014 to 2016 [28]. These authors found that both IMERG and TMPA overestimated light rainfall and underestimated large rainfall events, but IMERG performed better overall. Several studies have found TMPA and

CHIRPS to be comparable to direct rain gauge measurements in various regions including South America, North America, and Africa [29–31]. Extensive validation studies have been done with TMPA, but these studies have not been conducted over the LMRB or with as many rain gauge stations as in this study. There were no previous studies that evaluated CHIRPS for as many years or against as many in-situ stations in the LMRB as the methodology presented here. Using a more extensive in-situ data set with 477 stations from the Mekong River Commission, this study aims to closely analyze TMPA and CHIRPS over the 15-year period from 2000 to 2014. The time period for this study is based on the availability of rain gauge data and satellite-based sensor operation.

This research aims to determine the extent to which the satellite precipitation products TMPA and CHIRPS are able to estimate precipitation in the LMRB and, thus, show their validity for consideration in basin-scale water management decisions. Unlike previous studies in this region of the world, this research uses an extensive in-situ gauge network for validation, satellite estimates were compared against the rain gauge measurement(s) for the same pixel in order to assess the performance of the satellite products. Comparisons were performed based on classifications of the rain gauge locations with respect to rainfall accumulation and elevation to examine the extent to which the amount of rainfall and topography plays a role in their performance. Additionally, a spatial correlation analysis was applied to both the satellite-based products to visualize the geographical relationship with in-situ measurements and assess any spatial bias. The results of this validation study have the ability to improve estimation of water resources and benefit flood and drought forecasting systems in the LMRB by presenting the capabilities of TMPA and CHIRPS. [32]. It is important to note that the final goal of this work was not to estimate floods and extreme events, but to evaluate the performance of satellite precipitation estimates so that future studies can feel confident about using these estimates in their models. This study and further research applying in-situ observations to determine accuracy of satellite product can aid in the improvement of basin-wide decision-making, flood prediction, and management of floodwaters and drought by providing validations which suggest that satellite estimates can substitute for rain gauge measurements in areas with a sparse or absent in-situ network [26,33].

## 2. Materials and Methods

### 2.1. In-Situ Measurements

Daily precipitation totals were provided by the Mekong River Commission (MRC) from 481 in-situ rain gauge stations located throughout the basin from 1920 to 2014 (Figure 1) and were available upon request. For the time period selected in this study, 2000 to 2014, 477 stations in the LMRB had available precipitation measurements for this time period. The rain gauge data set contained gaps where no precipitation measurements were taken during the extended time periods for some stations. Specifically, 21% of the total amount of days from 2000 to 2014 for all rain gauge stations had unavailable or missing rainfall measurements. Gauges were not available consistently across the basin, significantly limiting data availability over large areas of the basin, as a result (Figure 1). Additionally, the quantity of rain gauges did not satisfy the size of the LMRB (1 station per 1580 km$^2$). Therefore, the data gaps and gauge sparsity in the LMRB make it impractical to use rain gauge data alone for hydrological decision making [34]. Here, the available in-situ data served as a validation dataset for evaluating the accuracy of the TMPA and CHIRPS satellite products.

### 2.2. Satellite Retrievals

Launched in 1997, NASA's TRMM Multi-Satellite Precipitation Analysis (TMPA) 3B42 v.7 is one of the most widely used satellite precipitation products and is very useful for hydrometeorological applications in data-sparse regions of the world [35]. TMPA combines information from the TRMM precipitation radar, passive microwave and infrared sensors from various satellites, and available rain gauge data to measure tropical rainfall for weather and climate research [19]. Monthly in-situ precipitation data were gathered from the Global Precipitation Climatology Project (GPCP) developed

by the Global Precipitation Climatological Center (GPCC) and the Climate Assessment and Monitoring System (CAMS) developed by NOAA's Climate Prediction Center (CPC), and were used for calibration of the TMPA product [36]. For a full explanation of the TMPA input datasets and algorithms, please refer to Huffman et al. (2007) [36]. Estimates were available at 3-hour intervals with 0.25° by 0.25° spatial resolution for the region 50°S to 50°N [36,37]. In this study, the final daily product (TRMM_3B42_Daily) derived from the 3-hourly estimate (TRMM_3B42) was used in the analysis.

Climate Hazards Group InfraRed Precipitation with Station data (CHIRPS) is a quasi-global precipitation product that provides estimates for over 30 years (1981 to near-present) and is provided by the Climate Hazards Center (CHC) [18]. CHIRPS uses a recently produced satellite rainfall algorithm that combines climatology data, satellite precipitation estimates, and in-situ rain-gauge measurements to produce a high resolution precipitation product [18]. It utilizes 0.05° satellite imagery alongside in-situ station data to produce a gridded rainfall product. CHIRPS is widely used for rainfall trend analysis and seasonal drought monitoring [19]. The climate data used in the CHIRPS methodology consists of two in-situ datasets, Agromet Group of the Food and Agriculture Organization of the United Nations (FAO) and Global Historical Climate Network (GHCN). These two data sets are long-term averages and were used to create the climate data used by CHIRPS. The station's historical data were mostly used in the calibration for the CHIRPS method instead of data from this study period, 2000–2014. It was also important to note that the CHIRPS methodology uses the TMPA product to calibrate global Cold Cloud Duration (CCD) precipitation estimates [19]. Although CHIRPS has a higher spatial resolution (0.05°) than TMPA (0.25°), this does not necessarily translate into a higher accuracy. However, higher spatial resolution helps in the characterization of the spatial variability. Here, daily estimates from CHIRPS (CHIRPS Daily Version 2.0 Final) were used in the analysis.

*2.3. Methodology*

In this study, rain gauge measurements provided by the Mekong River Commission were used as a validation dataset for two satellite-based precipitation products, TMPA and CHIRPS, in a point to pixel comparison via the methodology outlined in Figure 2. First, the measurement value of –9999 was removed from all in-situ data and those were treated as missing observations and were excluded from analyses. To match the daily satellite-based estimates of rainfall with the rain gauge measurements, the satellite pixel encompassing each rain gauge location was identified. In each pixel, we extracted the satellite-based estimate and paired it with the corresponding rain gauge data at a daily scale from 2000 to 2014. If more than one station was present within a satellite pixel, the rain gauge measurements were averaged before being compared to the satellite-based precipitation estimate in that pixel. Additionally, a validation study was employed to assess the satellite-based product's ability to correctly estimate precipitation (i.e., the rain–no-rain detection problem). This was done by determining the percentage that the satellite-based estimate and gauge measurement in a particular pixel were both wet (accumulating at least 0.06 mm of rainfall) or dry (below 0.06 mm of rainfall) and if one was wet and the other was dry.

Daily TMPA and CHIRPS estimates and in-situ measurements were separately aggregated to monthly and annual accumulation. Similarly, daily precipitation was aggregated to seasonal accumulation for comparison. The wet season included accumulations from June through September and the dry season included accumulations from November through February. Additionally, the rain gauges were categorized based on the average annual accumulation in the following ranges—0 to 1000 mm, 1001 mm to 1500 mm, 1501 mm to 2000 mm, 2001 mm to 2500 mm, and greater than 2500 mm. Classifying the rain gauges by monthly accumulation would reveal if and to what extent the amount of rainfall received could affect performance of the satellite-based product estimates. The rain gauges were also categorized based on elevation into the following ranges—0 to 100 m, 101 m to 300 m, 301 m to 500 m, 501 m to 1000 m, and greater than 1000 m. Classifying the rain gauges by elevation would determine the role topography plays in satellite-based product performance. For the monthly, annual, seasonal, and categorical analyses, dry days (days where the corresponding in-situ measurement and

satellite-based estimates were below the threshold of 0.06 mm) were excluded in order to evaluate only days where both the rain gauge and the satellite-based product indicated precipitation. However, the dry days were not excluded from the rain–no-rain validation study described previously so that the days where CHIRPS and TMPA correctly estimated no precipitation from a rain gauge measurement could be counted and evaluated.

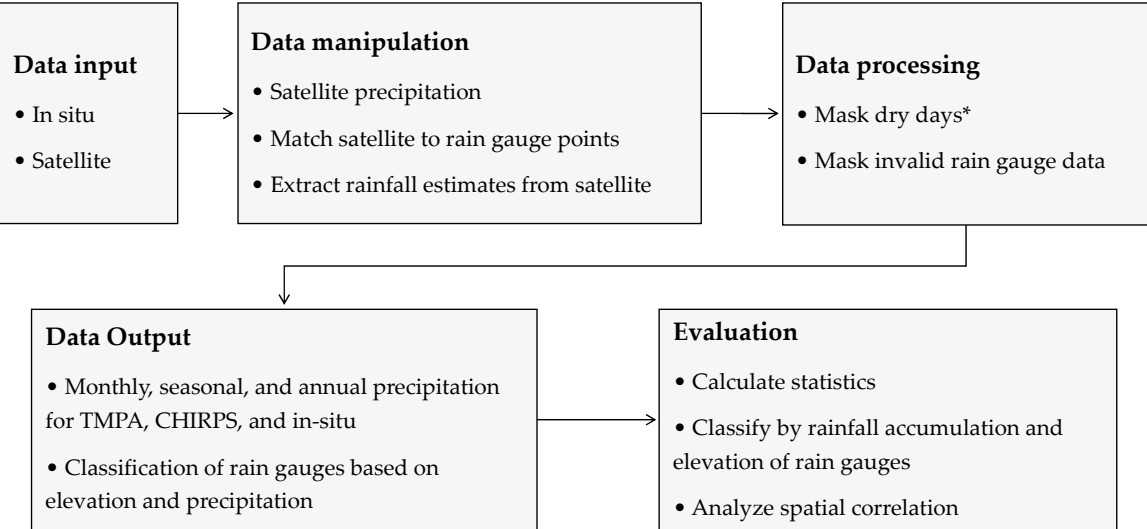

**Figure 2.** Methodology and workflow for this study. *Dry days are days where both satellite-based estimate and rain gauge measurement give a precipitation below the threshold of 0.06 mm. Dry days are excluded in some analyses but not all.

Several statistical metrics were employed for analyses between the satellite-based estimates and in-situ measurements. For the linear correlation analysis, the Pearson's correlation coefficient (r-value) was examined. The closer the r-value was to 1, the more highly correlated the satellite-based estimate was to the in-situ data. Bias was defined as the average of the difference between two quantities and showed the tendency of TMPA and CHIRPS to overestimate or underestimate corresponding in-situ measurements. The Root Mean Squared Error (RMSE) was used to assess the goodness of fit between the satellite-based and in-situ datasets. Lower RMSE values indicated a better fit between two variables, which here would imply a high correlation between satellite-based estimates and in-situ data. The Mean Absolute Error (MAE) provided the average of the absolute errors and measured the difference between two continuous variables, mean absolute error was defined as the average of all absolute differences between the two quantities. MAE provides further insight into the correlation between in-situ and satellite-based products. Each of these statistics was analyzed separately for the monthly, seasonal, and annual data, as well as the categorized data based on rain gauge elevation and rainfall accumulation.

## 3. Results

In the LMRB, there were large variations in precipitation intensity, duration, and accumulation particularly between the dry and wet seasons. We found that these variations were realized differently between the two satellite-based precipitation products, CHIRPS and TMPA. In-situ monthly rainfall accumulation ranged from 0.10 mm to 1748.0 mm, TMPA measured monthly rainfall from 0.09 mm to 1279.9 mm, and CHIRPS measured monthly rainfall from 0.06 mm to 1256.0 mm (Table S1). The dry season in-situ data produced anywhere from 0.30 mm to 612.07 mm per year were averaged over the 15-year study period (Table S1). On the other hand, TMPA recorded the dry season range to be from 0.17 mm to 292.85 mm and CHIRPS from 1.07 mm to 692.93 mm. The wet season produced rainfall accumulation from 170.25 mm to 2709.70 mm according to in-situ data, whereas TMPA estimated

a range from 86.46 mm to 1454.66 mm and CHIRPS estimated from 110.99 mm to 2138.24 mm (Table S1). In-situ cumulative annual rainfall ranged from 4.00 mm to 4551.50 mm, TMPA measured annual rainfall from 3.72 mm to 3029.82 mm, and CHIRPS measured annual rainfall from 1376.9 mm to 2136.4 mm (Table S1). While the elevation decreased from greater than 2000 m in the Northern reaches of the basin to the Southern Vietnam Delta at sea level, the precipitation followed an East to West gradient, with most rainfall accumulation in Vietnam, Laos, and Eastern Cambodia. Western parts of Thailand and Cambodia received the least amount of rainfall. Figure 3 shows the rainfall distribution patterns geographically for the dry, wet, and annual datasets derived from 2000 to 2014. Both CHIRPS and TMPA were able to represent the rainfall distribution over the basin in the dry season that was indicated by the rain gauges in Figure 3, although CHIRPS estimated much more precipitation in the Western part of the basin in Vietnam. A similar trend appeared in the wet season and the annual maps in Figure 3, such that TMPA and CHIRPS showed similar rainfall distributions over the LMRB with CHIRPS having slightly higher estimations. The annual distribution was very similar, aside from the satellite overestimating gauge measurements in several areas. Satellite-based estimates showed higher correlation with rain gauge measurements during the dry season and lower correlation during the wet season where the in-situ data recorded much lower rainfall accumulation than the satellite-based estimation. Table 1 explains the rain–no-rain detection accuracy by CHIRPS and TMPA, when compared to the rainfall recorded by rain gauges. CHIRPS correctly detected rain 21.9% of the time and TMPA correctly detected rain 15.7% of the time, compared to the daily rainfall. CHIRPS agreed with the in-situ for no-rain days 44.9% of the time and TMPA agreed with in-situ for no-rain days 49.1% of the time. However, both TMPA and CHIRPS estimated rain for more than 20% of the days when the rain gauges did not measure any precipitation. CHIRPS agreed with in-situ 66.8% for rain or no-rain days and TMPA agreed with in-situ 64.9% for rain or no-rain days (Table 1). The validation study showed that CHIRPS was better able to estimate whether a dry or wet day was present.

For further analysis, the rain gauges were classified by the following ranges for annual accumulation—0 to 1000 mm, 1001 mm to 1500 mm, 1501 mm to 2000 mm, 2001 mm to 2500 mm, and greater than 2500 mm. Table 2 shows the analysis based on monthly rainfall accumulation, which analyzes the r-value, bias, MAE, and RMSE. For stations receiving more than 2500 mm of annual rainfall, CHIRPS had an r-value of 0.83 and TMPA had an r-value of 0.65. Both CHIRPS and TMPA had better correlation with each subsequent accumulation category, indicating that the satellite-based products performed better in areas with high precipitation. There was no apparent relationship between the number of stations in each category in Table 2 and the correlation between in-situ and satellite-based estimates. CHIRPS detected rainfall in each class significantly better than TMPA. To visualize the monthly correlations, Figure 4 shows a side-by-side boxplot comparison between the satellite-based and in-situ measurements of the average monthly rainfall accumulation from 2000 to 2014. The seasonal data sets used in this study were configured from this plot using the four highest precipitation months (June, July, August, and September) for the wet season and the four lowest precipitation months (November, December, January, February) for the dry season. The satellite-based estimates overestimated rain gauges during the peak of the wet season (July and August) (Figure 4). Further, TMPA and CHIRPS both recorded July as the peak of the wet season (the month with the highest average rainfall accumulation), whereas the rain gauges showed August to be the peak of the wet season. Similarly, TMPA and CHIRPS indicated January as the lowest accumulation of precipitation, whereas the rain gauges showed February to be the month with lowest rainfall accumulation. In each of the three boxplots, October had the highest variance in cumulative precipitation from 2000 to 2014, denoted by the largest vertical black bars. To this end, the months receiving lower rainfall amounts showed higher correlation between the satellite-based and in-situ measurements than the months receiving higher rainfall amounts. Figure 5 shows the time-series trend from the satellite-based estimates and provides a closer look of comparison on a monthly scale.

From the time-series we conclude that the estimates from both CHIRPS and TMPA were closely correlated with the seasonal patterns of the in-situ measurements (Figure 5).

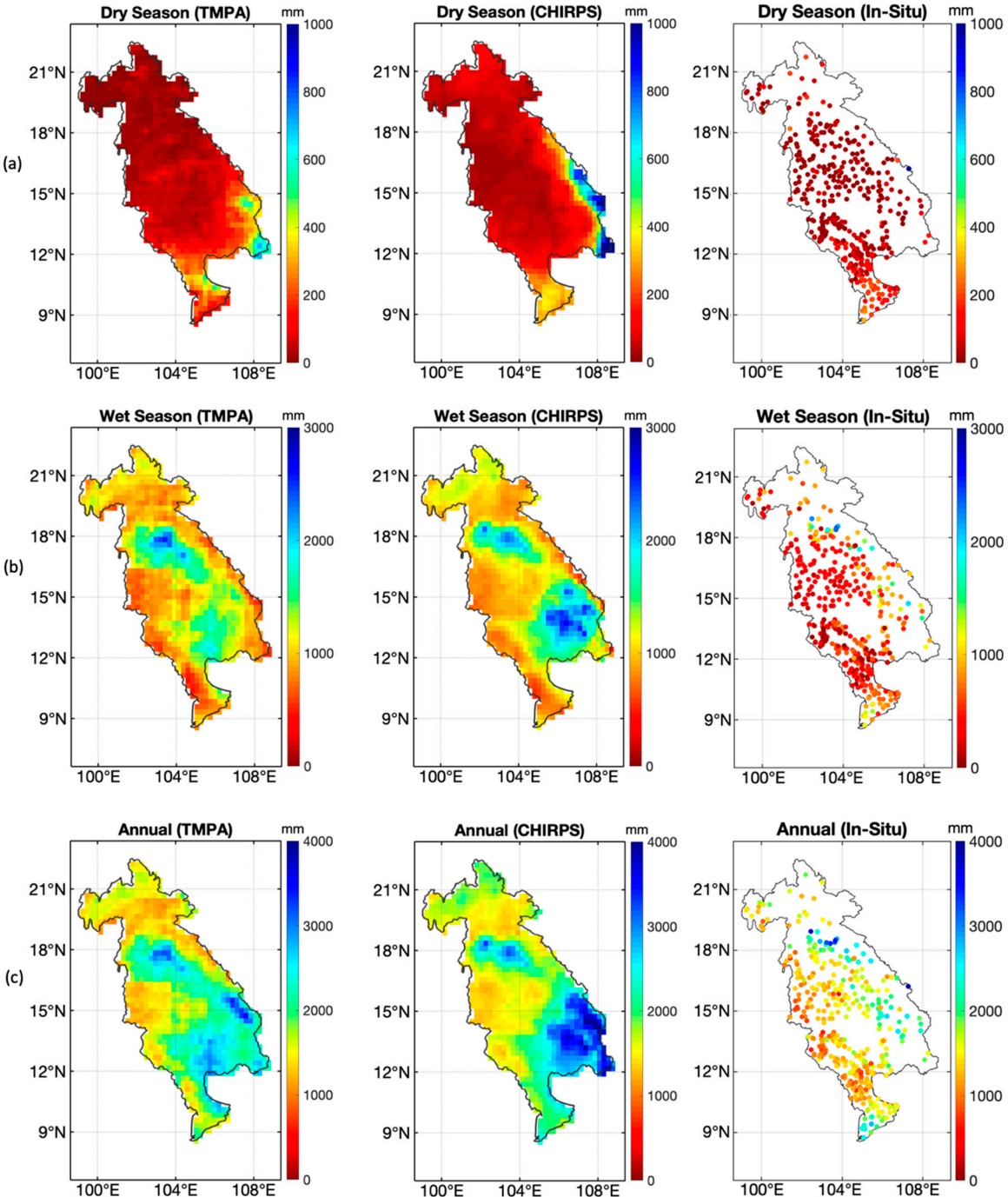

**Figure 3.** Rainfall Distribution in Lower Mekong River Basin. (**a**) Average (average over 2000 to 2014) seasonal rainfall accumulation during the dry season (November to February) for Tropical Rainfall Measuring Mission (TRMM) Multi-Satellite Precipitation Analysis (TMPA), Climate Hazards Group InfraRed Precipitation with Station (CHIRPS), and in-situ. (**b**) Average seasonal rainfall accumulation during the wet season (June to September) for TMPA, CHIRPS, and in-situ. (**c**) Average annual rainfall accumulation for TMPA, CHIRPS, and in-situ.

**Table 1.** Rain–no-rain validation study results for daily precipitation from 2000–2014.

|  | CHIRPS - No Rain | CHIRPS - Rain |
|---|---|---|
| **Rain Gauge - No Rain** | 44.9% | 26.4% |
| **Rain Gauge - Rain** | 6.8% | 21.9% |
|  | **TMPA - No Rain** | **TMPA - Rain** |
| **Rain Gauge - No Rain** | 49.1% | 22.3% |
| **Rain Gauge - Rain** | 13.0% | 15.7% |
|  | **Correct Detection** | **Incorrect Detection** |
| **CHIRPS** | 66.8% | 33.2% |
| **TMPA** | 64.9% | 35.1% |

**Table 2.** Comparison statistic results between rain gauge measurements and satellite-based precipitation estimates for monthly rainfall classes based on in situ accumulation from 2000 to 2014 for R, bias, mean absolute error (MAE), and RMSE.

| In-Situ Annual Accumulation | Satellite-Based Product | r-value | Bias (mm) | MAE (mm) | RMSE (mm) |
|---|---|---|---|---|---|
| 0 – 1000 mm **(43 stations)** | CHIRPS | 0.72 | −31.28 | 43.44 | 63.45 |
|  | TMPA | 0.51 | 5.55 | 43.78 | 65.88 |
| 1001 – 1500 mm **(207 stations)** | CHIRPS | 0.75 | −36.28 | 52.52 | 76.49 |
|  | TMPA | 0.56 | 9.82 | 59.48 | 91.72 |
| 1501 – 2000 mm **(124 stations)** | CHIRPS | 0.79 | −33.52 | 58.23 | 87.77 |
|  | TMPA | 0.61 | 0.85 | 67.74 | 103.98 |
| 2001 – 2500 mm **(41 stations)** | CHIRPS | 0.82 | −31.01 | 72.28 | 117.38 |
|  | TMPA | 0.64 | −29.10 | 84.29 | 136.75 |
| > 2500 mm **(21 stations)** | CHIRPS | 0.83 | −46.59 | 97.78 | 161.46 |
|  | TMPA | 0.65 | −88.51 | 129.40 | 208.06 |

For further comparison of the data products, the correlation coefficient, bias, MAE, and RMSE were determined. During the wet season, TMPA underestimated the rain gauge measurements more than during the dry season. The correlation coefficient (r-value) between the rain gauge and TMPA estimates was 0.38 for dry the season comparison, 0.48 for the wet season comparison, 0.49 for the annual comparison (Table S2). The CHIRPS comparison showed an r-value of 0.61 for the dry season comparison, 0.68 for the wet season comparison, and 0.58 for the annual comparison (Table S3). The average MAE for the comparison between in-situ and TMPA was 0.07 mm for the dry season comparison, −1.26 mm for the annual comparison, and −17.37 mm for the wet season comparison (Table S2). The comparison between CHIRPS and in-situ produced average MAE values of −2.81 mm for the dry season comparison, −23.91 mm for the annual comparison, and −162.46 mm for the wet season comparison (Table S3). Overall, TMPA and CHIRPS both correlated better during the wet seasons than the dry seasons most likely due to the low variance in the estimates from the dry season. The correlation between TMPA satellite-based estimates and rain gauge measurements in this study was ordered (from most correlation to least correlation) as follows—annual comparison, wet season comparison, monthly comparison, and lastly dry season comparison. However, CHIRPS correlated as follows—wet season, dry season, annual, and monthly.

Table 3 shows that the rain gauges were categorized based on their elevation and were used for analysis at different rainfall accumulation classes. They were classified into the following categories—0 to 100 m, 101 m to 300 m, 301 m to 500 m, 501 m to 1000 m, and greater than 1000 m. From this table, we conclude that neither CHIRPS nor TMPA were significantly impacted by elevation nor was the

correlation affected by the number of stations in each category. CHIRPS was more highly correlated to in situ measurements for rain gauges at elevations between 101 m and 300 m, with an average r-value of 0.84 for these stations. TMPA also showed a better agreement with the in-situ data in this elevation category, but had an r-value of 0.69. At stations with elevations above 1000 m, CHIRPS performed much better than TMPA, having an r-value of 0.81 as compared to TMPA which had an r-value of 0.54. It is important to note that as the elevation range increased, the number of rain gauges that fall within the subsequent category decreased. This could have affected the results of the study since each elevation category had unequal number of rain gage stations.

For spatial correlation analysis, the r-value at each station was determined based on the monthly rainfall accumulation and was plotted at each station location. The r-values could be visualized for CHIRPS and TMPA in Figures 6 and 7, respectively. When comparing these two figures, it was apparent that CHIRPS was more highly correlated for the majority of all rain gauge stations across the basin than TMPA. CHIRPS did not display a distinct spatial pattern of correlation (Figure 6). Most stations had an r-value between 0.6 and 1.0 and were distributed widely across the basin. There were only few stations with r-values between 0.4 and 0.6 in the areas of Cambodia and no stations with r-values between 0.0 and 0.4. CHIRPS had the highest correlation in the northern and central areas and the Vietnam delta and the least correlation in the southwestern regions of Cambodia. With significantly less r-values, the TMPA analysis showed more of a spatial pattern of correlation than the CHIRPS spatial analysis (Figure 7). Stations with r-values between 0.8 and 1.0 were mostly collected in the central region of the basin in Eastern Thailand and Vietnam and represented the highest correlation between TMPA and in-situ measurements. Overall, CHIRPS had higher r-values than TMPA for the monthly rainfall spatial comparison, which indicated that CHIRPS might be able to better spatially represent precipitation in the LMRB.

**Table 3.** Comparison statistic results between rain gauge measurements and satellite-based precipitation estimates for monthly rainfall categorized on the basis of elevation for R, bias, MAE, and RMSE.

| Rain Gauge Elevation | Satellite-Based Product | r-value | Bias (mm) | MAE (mm) | RMSE (mm) |
|---|---|---|---|---|---|
| 0 – 100 m (240 stations) | CHIRPS | 0.79 | −36.91 | 56.71 | 85.41 |
| | TMPA | 0.54 | 15.52 | 66.36 | 107.11 |
| 101 – 300 m (152 stations) | CHIRPS | 0.84 | −42.82 | 63.03 | 101.58 |
| | TMPA | 0.69 | −33.49 | 69.50 | 111.71 |
| 301 – 500 m (18 stations) | CHIRPS | 0.80 | −15.07 | 59.57 | 98.56 |
| | TMPA | 0.63 | −19.79 | 70.62 | 119.10 |
| 501 – 1000 m (38 stations) | CHIRPS | 0.81 | −5.41 | 57.69 | 95.20 |
| | TMPA | 0.59 | −8.69 | 73.25 | 120.21 |
| >1000 m (12 stations) | CHIRPS | 0.81 | −22.47 | 56.99 | 103.75 |
| | TMPA | 0.54 | 23.35 | 92.33 | 152.24 |

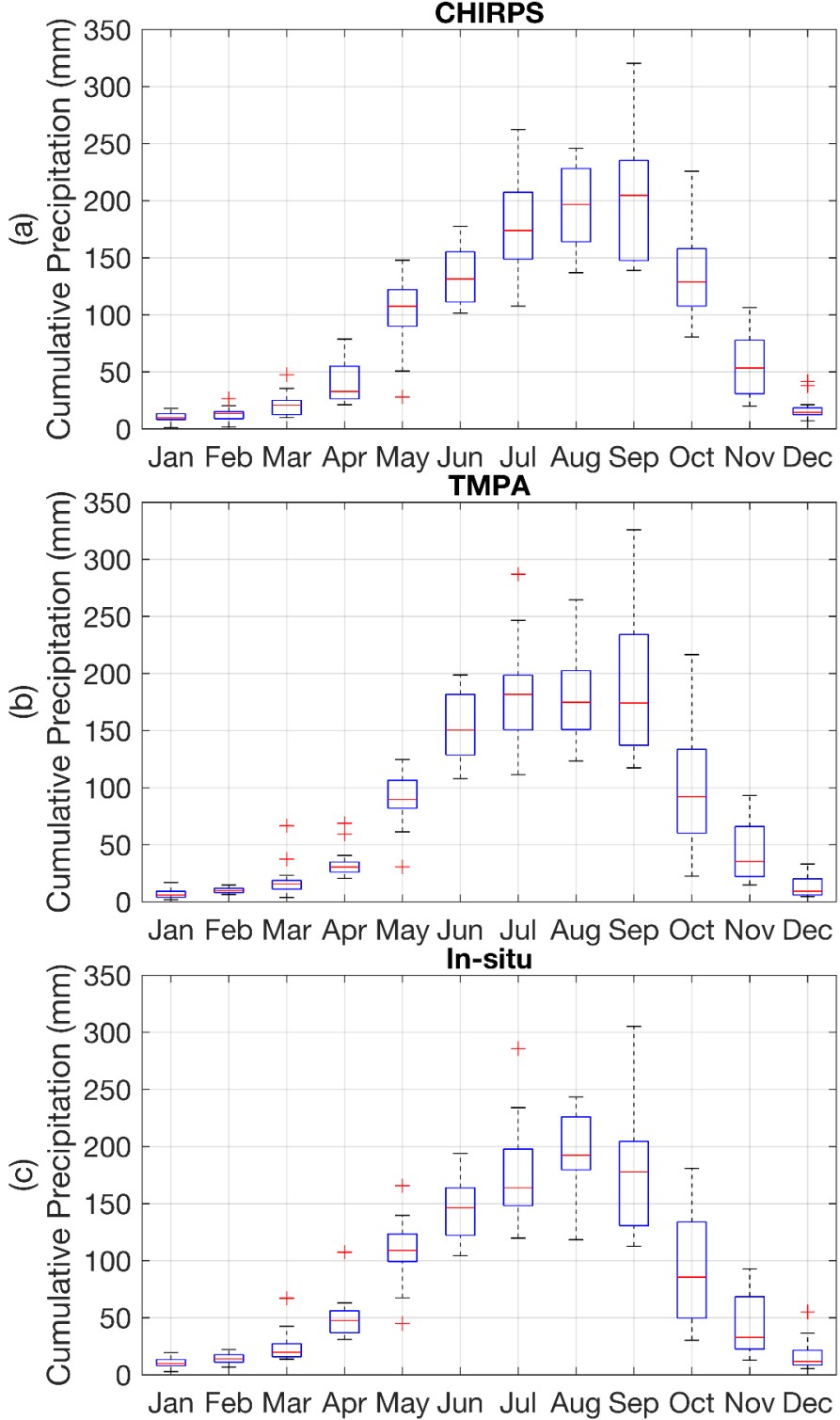

**Figure 4.** (**a**) Boxplot analysis of each month using rain gauge measurements. (**b**) Boxplot analysis of each month using TMPA estimates. (**c**) Boxplot analysis of each month using CHIRPS estimates. Each analysis uses data from 2000 to 2014. Red horizontal bars represent the median rainfall amount. The blue boxes represent the data that is within the 25th and 75th percentiles. The black horizontal bars above and below the blue boxes represent the maximum and minimum rainfall amounts, respectively. The red '+' represent outliers in the data set.

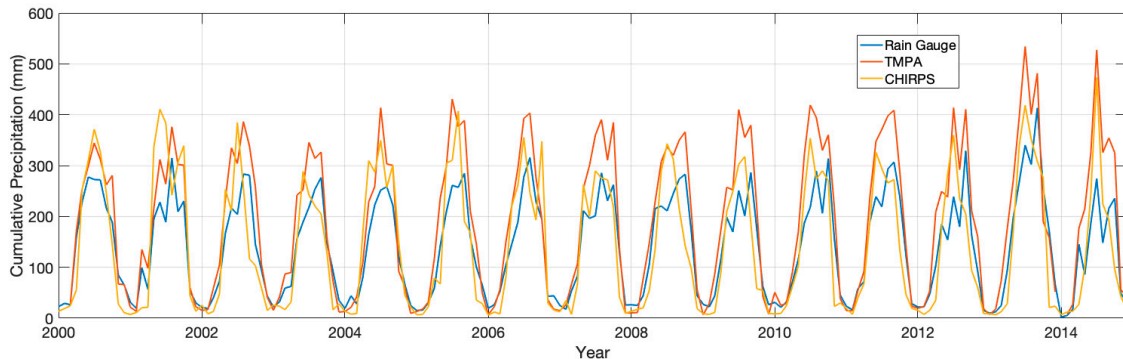

**Figure 5.** Time-series comparison of monthly averages from in-situ data and TMPA and CHIRPS satellite-based precipitation estimates from 2000 to 2014.

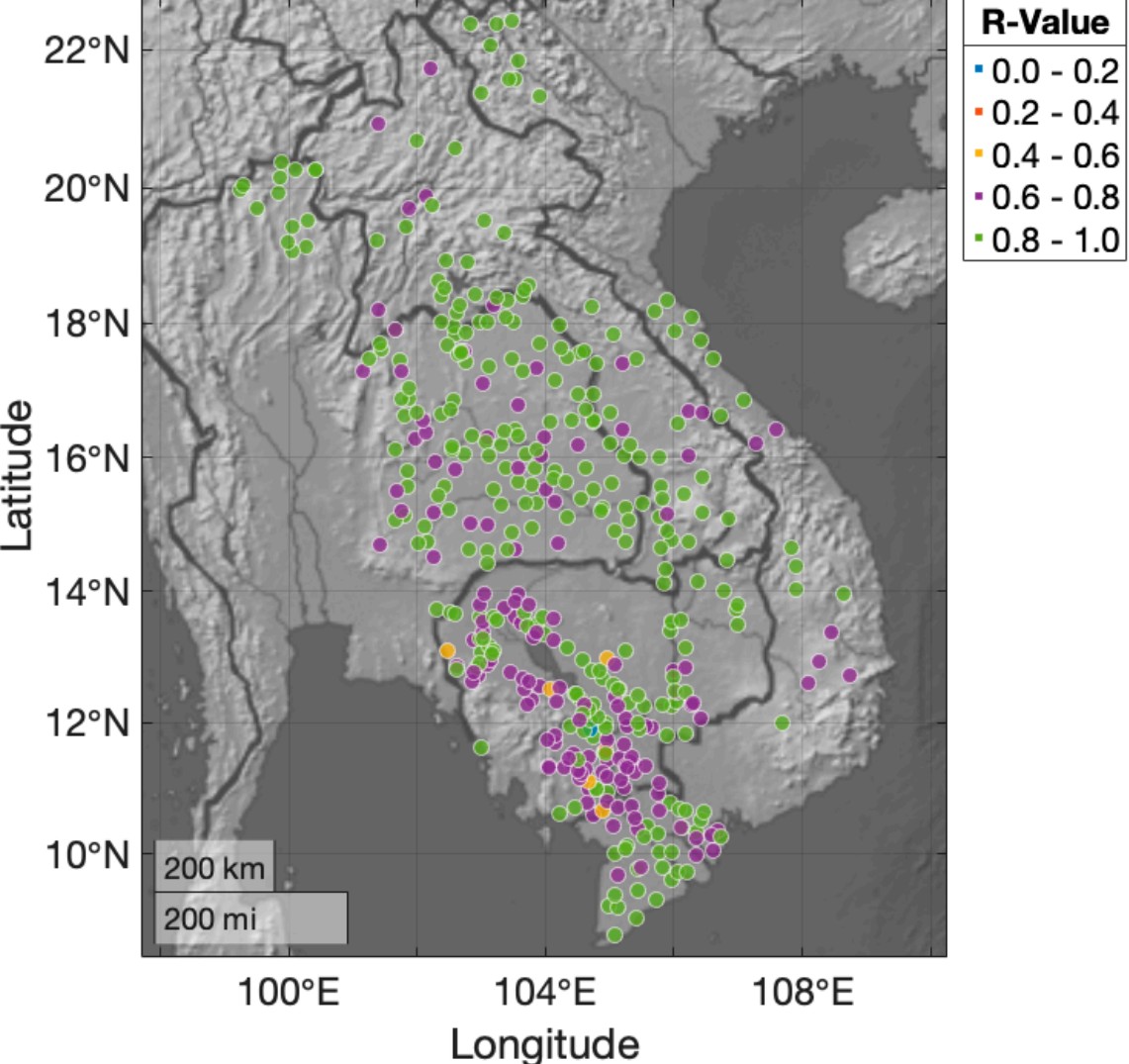

**Figure 6.** Spatial correlation results of mean r-value for each rain gauge stations against CHIRPS precipitation estimates for monthly rainfall.

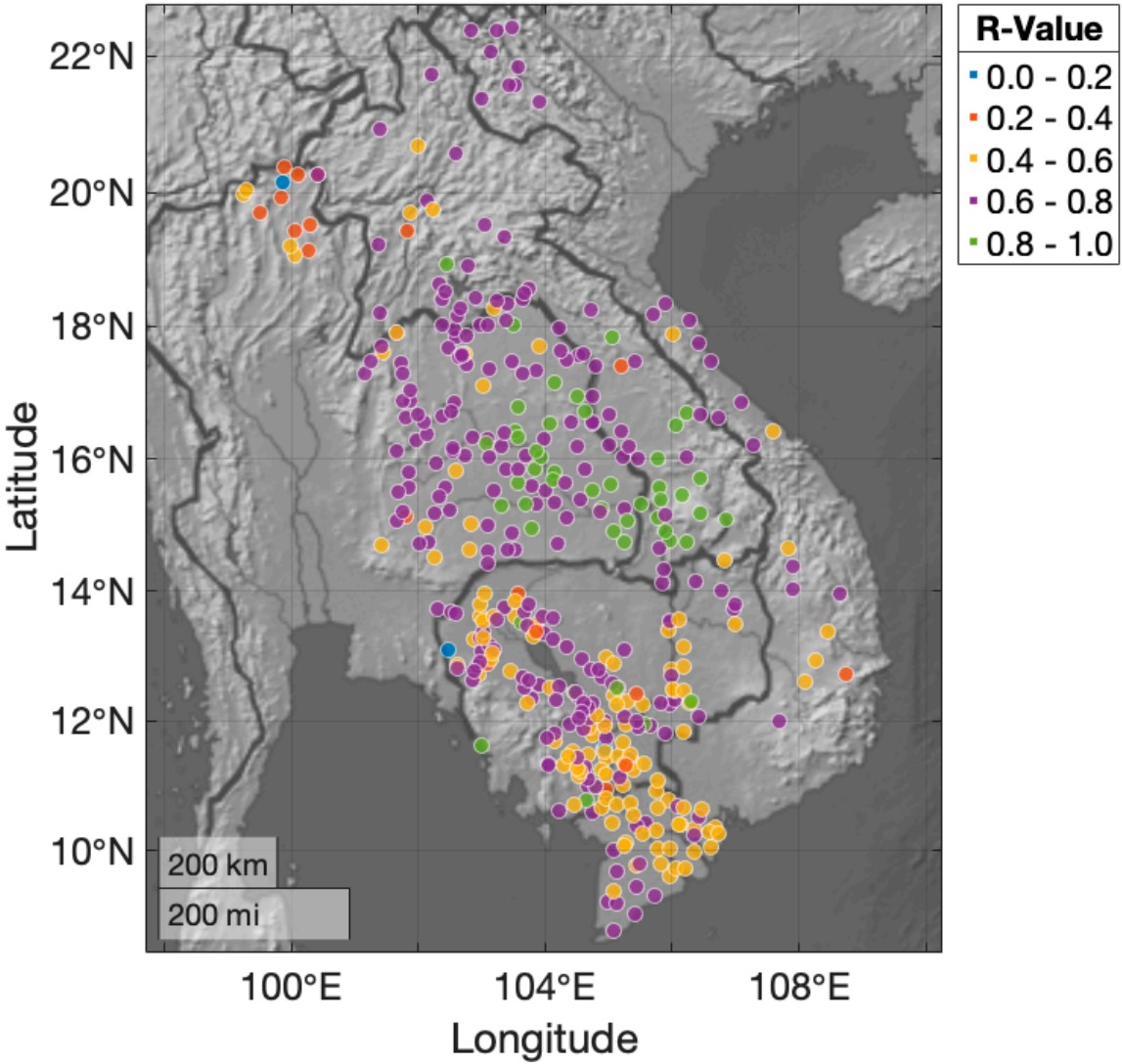

**Figure 7.** Spatial correlation results of mean r-value for each rain gauge stations against TMPA precipitation estimates for monthly rainfall.

## 4. Discussion

This study analyzed the performance of NASA's TMPA satellite-based precipitation estimates (3B42 v.7) and Climate Hazards Group InfraRed Precipitation with Station data (CHIRPS) in the Lower Mekong River Basin using an extensive rain gauge network for validation. A detailed comparison was performed between the satellite-based rainfall products and in-situ measurements from 477 rain gauge stations. A rain–no-rain detection analysis showed that TMPA made a correct detection of a wet or dry day 64.99% of total days and CHIRPS made a correct detection of a wet or dry day 66.8% of total days, when compared to all in-situ daily measurements. With this validation study, we conclude that CHIRPS was better able to distinguish wet and dry days than TMPA. Satellite-based rainfall estimates were compared monthly, seasonally, and annually to in-situ data for the time period from 2000 to 2014. Additionally, rain gauges were categorized on the basis of elevation and mean rainfall accumulation and were compared to the corresponding TMPA and CHIRPS pixels to determine the effects of topography and amount of rainfall on satellite-based product estimation. When averaged over the entire LMRB, the satellite-based data mostly retained the overall annual precipitation patterns and geographic distributions. Overall, TMPA overestimated in-situ rainfall in the dry seasons, whereas CHIRPS underestimated rainfall in the dry seasons. Furthermore,

both satellite-based estimates were more highly correlated to in-situ data during the wet season (June–September) than dry season (November–February). Similarly, the annual comparisons between in-situ and both satellite-based estimates showed higher correlations than the monthly comparisons when analyzed over the fifteen-year study period. The bias and false detections in the satellite-based estimates could be caused by topography, rain gauge data availability, or amount of precipitation received in certain locations. The r-values were determined at each station location based on monthly rainfall accumulation from in-situ measurements and both satellite-based products. CHIRPS was more highly correlated to the rain gauge stations across the basin than TMPA, with the majority of stations having r-values of 0.8 to 1.0, when compared to CHIRPS. Additionally, CHIRPS did not show a distinct spatial pattern of correlation, whereas TMPA did show a geographical pattern. The correlation dependence on geography and climate could be explained by TMPA being more affected by the annual monsoon movement. Overall, the spatial comparison showed CHIRPS to have a higher correlation than TMPA with rain gauge measurements in the LMRB, which indicated that CHIRPS might be able to better spatially represent rainfall.

As stated previously, other precipitation comparisons have been implemented and vary by location, number of rain gauges, satellite-based product, and study period. The results of these studies also differ in whether satellite-based estimates overestimated or underestimated in-situ measurements. A study by Katsanos et al. (2004), found a higher bias in the satellite-based estimates during peak precipitation periods, and this study also found high biases during peaks in the wet season [38]. For example, the results from this comparison were similar to a study by Collischonn et al. (2008) over the Tapajos River Basin in Brazil, in which TMPA estimates were found to be very close to the in-situ measurements when averaged over the entire river basin and that TMPA mostly underestimated precipitation in their study [39]. Additionally, a study by Su et al. (2008) over the La Plata Basin in South America found TMPA to be less accurate during high rain rates at a daily time scale and to overestimate rainfall, which was similar to the results of this study during the wet season that was represented by high rain rates [40]. In their study, TMPA was able to represent low flows but had a positive bias during peak flows in satellite-driven model simulations. In order to use TMPA in the LMRB for similar satellite-driven watershed modeling, such biases would need to be accounted for and adjusted to more accurately estimate streamflow and capture flooding events. After comparing TMPA and CHIRPS to rain gauge measurements, the results of this research showed that CHIRPS might be better at representing precipitation in the LMRB than TMPA. However, Xian et al. (2018) found TMPA to be superior to CHIRPS in hydrological simulation using SWAT [21]. Furthermore, spatial resolution did play a role in the validation of these precipitation products. CHIRPS had a spatial resolution of 0.05° and TMPA had a resolution of 0.25°. Generally, higher spatial resolution translates to higher accuracy, but this was dependent on the method used to generate this product. What we imply is an inferior method used to generate a high spatial resolution product that might have a lower accuracy than a superior method used to generate a lower spatial resolution product. In this study we find that the accuracy of TMPA and CHIRPS were very close, but the higher spatial resolution of CHIRPS might provide an advantage in the accuracy when compared to rain gauges.

## 5. Conclusions

This work was one of the first attempts at evaluating the satellite-based precipitation data products in the Lower Mekong River Basin with such an extensive in-situ dataset. The hydrologic significance of TMPA and CHIRPS in the LMRB could be assessed from the results of this study and other analogous validation studies. In addition, a similar methodology to the one described here could be applied to the GPM IMERG data to further assess the performance of satellite-based precipitation products in the region. The important broad impacts of this research are the implications of remotely sensed products in hydrologic cycle modeling, specifically in the LMRB or similar un-gauged basins. For future validation studies of satellite-based estimates, this methodology could be applied to new, higher resolution products like GPM IMERG, to look at the progression and advancement in satellite-based estimation.

With better temporal and spatial coverage, satellite-based inputs will serve as an improvement, compared to precipitation from rain gauges for various modeling in basins like LMRB where there is a sparse coverage of rain gauges. Additionally, evaluation of satellite-based products is essential for improvement upon satellite-based algorithms and equipment [41]. Given the observed increase in accuracy of remotely sensed precipitation products (sensor configurations, improved spatial resolution, and temporal repeat), a careful comparison of the fidelity of each product, as shown here, is helpful for assessing their utility for basin-scale modeling capabilities, particularly for water resource management applications in poorly-gauged basins such as LMRB. This study undertook a unique approach at comparing TMPA and CHIRPS estimates with in-situ observations in the LMRB. We conclude that precipitation from TMPA and CHIRPS could be used reliably in hydrological applications in rain gauge sparse regions of the world.

**Supplementary Materials:** The following are available online at http://www.mdpi.com/2072-4292/11/22/2709/s1, Table S1: Minimum, mean, and maximum monthly precipitation accumulations and the standard deviation for monthly, dry, wet, and annual data sets for in-situ and satellite-based estimates, Table S2: Comparison statistic results between rain gauge measurements and TMPA 3B42 v.7 satellite-based estimates for the dry season, wet season, and annually from 2000 to 2014 for mean R, bias, MAE, and RMSE, Table S3: Comparison statistic results between rain gauge measurements and CHIRPS satellite-based estimates for the dry season, wet season, and annually from 2000 to 2014 for mean r-value, bias, MAE, and RMSE.

**Author Contributions:** Conceptualization, C.D. and V.L.; methodology, C.D.; software, V.L.; validation, C.D., V.L. and J.B.; formal analysis, C.D.; investigation, C.D.; resources, V.L.; data curation, C.D. and V.L.; writing—original draft preparation, C.D.; writing—review and editing, V.L., J.B. and R.S.; visualization, C.D. and V.L.; supervision, V.L., J.B., and R.S.; project administration, V.L.; funding acquisition, V.L.

**Funding:** This research received no external funding.

**Acknowledgments:** The authors wish to acknowledge the support from NASA SERVIR Program contract number NNX16AT86G "Improved Hydrologic Decision Support for the Lower Mekong River Basin Through Integrated Remote Sensing and Modeling."

**Conflicts of Interest:** The authors declare no conflict of interest.

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
