# Peer review of "Evaluation of Satellite-Based Rainfall Estimates in the Lower Mekong River Basin (Southeast Asia)"

_remotesensing, doi:10.3390/rs11222709_

Round 1

Reviewer 1 Report

The manuscript “Evaluation of Satellite-Based Rainfall Estimates in the Lower Mekong River Basin” is within the scope of the journal and addresses a relevant topic of interest to a broad audience.

The authors validated TMPA and CHIRPS products using a large set of in situ rainfall gauges. Analysis concerning the performance of CHIRPS are relatively more important considering the abundance of TMPA validation works available, including for SE Asia.

The introduction is somewhat general, offering little to differentiate it from previous works on the subject.

The authors should provide more information on the quality of the in situ dataset. The authors mention gaps and problems but are never specific. In the end, the reader is unable to understand what’s the percentage of measurements per station available. This can have an important impact on the outcome of the study and should be carefully presented. Are there cells with more than one gauge? Were there attempts at aggregating the gauges?

It is somewhat difficult to understand why the analysis was halted in 2014. This precludes the analysis of newer products. Considering the authors mention the applications of the datasets used in the study, including regional water resources management, this choice should be better explained.

The Pearson Correlation Coefficient should be referred to as r and not R. The statistical analysis is very simple and doesn’t go beyond what the simpler papers in the field cover.

The Discussion and Conclusions section is highly insufficient. The section should be subdivided and the discussion much improved. Although the authors include maps with the spatial distribution of r values for both products, they are not carefully discussed (in fact almost not discussed at all).

The abundance of data would justify (and call for) a detailed analysis of the performance results, in light of regional climate patterns and possible applications. This is absent of the manuscript.

The number of figures is excessive. Some of the figures could probably be included as supplementary material.

In summary, although the manuscript has a lot of potential, the authors fail to deliver it. As such, I cannot recommend it for publication at this time, unless revised significantly.

Reviewer 2 Report

The paper evaluates two satellite products, TMPA and CHIRPS at Lower Mekong River Basin (LMRB). This study is complementing several other studies in this region by inclusion of extensive gauge network. The authors find that CHIRPS show better performance than TMPA in the region. However, the products have different resolution. It would be helpful to include additional satellite products of same resolution. Some revisions are required before it’s acceptance for publication.

Following are the suggestions:

Line 102: Authors discussed the study by Guo et al. (2017) on the LMRB basin using only 38 stations. Add what criteria was used by Guo et al. to select only these few stations.

Line 117: In addition to the references, add the name of the regions.

Line 143: What is meant by “valid precipitation measurements”? Clarify the selection method of the rain gauges. How many stations out of 477 have missing data as the authors suggest “many stations”? Add a figure showing the annual average rainfall over the study area and what is the mean annual, wet, dry rainfall during the study period?

Line 195: The authors stated, “Dry days are excluded in some analyses but not all”. Explain where and why these were excluded.

Lines 227-228: What do the authors mean by “these variations are realized differently in the in-situ data”? Gauge only records observed data.

Lines 241-244: What is meant by “similar” and “dissimilar”?

Line 325: The results of elevation category could be impacted from the location of the gauges. The rainfall characteristics may not be same in the entire region. Discuss the rainfall patterns in different part of the region.

Line 386: The authors suggest that “bias and false detection … by topography, …”. In lines 315-318, authors present the results showing that neither of the satellite products are not significantly impacted by elevation. The authors need to add explanation of impacts of topography further. It would be helpful to have a figure which shows the number of gages per elevation bands within the watershed. Also add additional columns in Table 6 to include false and correct detection of rainfall per elevation band.

Reviewer 3 Report

This study examines the suitability of satellite data-based TMPA and CHIRPS precipitation products for precipiation estimates in the Lower Mekong River basin by comparing these products with in-situ rain gauge measurements. This study is well-structured and also reads well. I have no major objections, but some minor to moderate comments and suggestions:

L48: ‚precipitation results in drought‘ – please consider reformulation

L74: ground radar data are not available or not used (why)?

Fig. 1: Cambodia and Lausa are hard to read, 370 km scale is weird; please consider unificatin of the graphical style with Figs. 6 and 7

L86: please state who is a provide of CHIRPS data?

L99-120: this information might be better readable as a table?

L141: please refer to these data properly – are freely available or available on request?

L141-142: what is the density of rainfall gauges (e.g. stations per 1,000 km2)?

L143-145: please refer to the section where you describe how missing data were treated

L154: does that mean that TMPA product is already validated with rain gauge data? Please discuss how can this influence obtained results (corellations)?; the same for CHIRPS (L165)

L180: how many (%) of -9999 values did you have in your dataset? What is the spatio-temporal distribution of missing data and how they could influence obtained results?

L190: what threshold?

L248-249: these figures (21.9% and 15.7%) are bit confusing; for the first reading I had a feeling that only 21.9% rainy days were detected as rainy days, before I realised that these are 21.9% of all days; instead, it might be more understandable to calculate what % of rainy days were correctly detected as rainy days, etc.

Fig. 3: for easier comparison, please consider interpolation of in situ data; with that, please consider difference maps showing positive and negative differences between satellite products and in situ data

L299-310: better fitting in discussion section?

L333-338: better fitting in discussion section?

Tab. 4: please consider merging with Tab. 5

L371: see my previous comment to these figures

L378-380: this is not completely true for TMPA which exhibits close-to-zero biases for Jun and July and even positive bias for September (see Tab. 4); please consider reformulation

L394-397: please check the use of ‚this study‘ in this sentence

L404-406: yes, but the negative bias value for the wet season is not so strong compared to dry season (Tab. 5)

L406-408: which data supports this claim?

I’ll be happy to see this interesting work being published, but I recommend some moderate revisions of the manuscript to be done first.

Reviewer 4 Report

Please, see the enclosed file.

Author Response

This manuscript is a resubmission of an earlier submission. The following is a list of the peer review reports and author responses from that submission.

Round 1

Reviewer 1 Report

The manuscript “Evaluation of Satellite-Based Rainfall Estimates in the Lower Mekong River Basin” is within the scope of the journal and addresses a relevant topic of interest to a broad audience.

The abstract is adequate, but could be more informative and include some additional information on the performance of the products.

The introduction could be more concise but include more information on previous results. The authors mention several works addressing the validation of rainfall estimates but provide little detail about them and why an additional study is needed.

The in situ data should be explained in greater detail, including the type of gauges available, quality control steps and amount of valid data.

Correct ‘weighs’, Line 190.

Why didn’t the authors calculate the rain/no-rain metrics used to populate contingency tables. These allow the calculation of relevant metrics like POD, FBI, etc…

It would also be relevant to evaluate different temporal aggregations, from daily to monthly.

The evaluated products performed rather poorly for monthly fields. It would be important for the authors to provide a careful analysis of the results, integrating the output of their analysis with regional rainfall patterns, topography, etc

The manuscript provides a relevant contribution to the validation of rainfall estimates, for a part of the world where such data are necessary. However, the manuscript can be improved prior to publication. It should be more concise and the message streamlined, with greater integration of the results with previous results and geographic data. However, I strongly encourage the authors to resubmit a revised version of the manuscript.

Reviewer 2 Report

Major comments

This manuscript, entitled “Evaluation of Satellite-Based Rainfall Estimates in the Lower Mekong River Basin” by Chelsea Dandridge et al. compared TRMM Multi-Satellite Precipitation Analysis 3B42 v.7 and Climate Hazards Group InfraRed Precipitation with Station data (CHIRPS) daily rainfall estimates against rain gauge observations from 2000 to 2014, in the Lower Mekong River Basin (LMRB) in Southeast Asia.

This manuscript does not provide enough new information overall, in terms of accuracy of satellite rainfall estimates. The observation and interpolation methods (kriging method) are inadequate. I believe that the manuscript is unsuitable for publication in its current form, and requires substantial revisions and additional thorough analyses prior to being considered for acceptance. My primary concerns are the following:

(1) The interpolation method for developing gridded precipitation from an in-situ precipitation gauge network is not clear. Please explain the grid size of interpolated in-situ precipitation in LMRB.   In addition, the authors should estimate the accuracy of the kriging interpolation method. For example, they can evaluate the accuracy of kriging interpolation methodology following Liston and Elder (2006) and Suzuki et al. (2011). The accuracy of the interpolation method was evaluated with and without specific observation points, in which case the uncertainty of the interpolation method should have been noted.

Please check whether such an uncertainty range is smaller than the discrepancy between the basin in-situ observation-based averaged precipitation and TRMM or CHIRPS datasets.

Reference:

Liston, G.E., Elder, K., 2006. A Meteorological Distribution System for High-Resolution Terrestrial Modeling (MicroMet). J. Hydrometeor 7, 217–234. doi:10.1175/JHM486.1 Suzuki, K., Y. Kodama, T. Nakai, G. E. Liston, K. Yamamoto, T. Ohata, Y. Ishii, A. Sumida, T. Hara, and T. Ohta (2011) Impact of land-use changes on snow in a forested region with heavy snowfall in Hokkaido, Japan. Hydrological Sciences Journal-Journal Des Sciences Hydrologiques, 56(3), 443-467.

(2) It is not adequate to evaluate the accuracy of the spatial pattern of daily precipitation datasets between products. It is not interesting to compare seasonal and annual precipitation using the basin-averaged precipitation. The authors can calculate the spatial correlation between TRMM or CHIRPS and in-situ precipitation. Describing such correlations can be more important for estimating river runoff from LMRB by indicating the geographical precipitation consistency between satellite-based and in-situ-based precipitation.      

(3) If the authors' final goal is to estimate floods and extreme events in LMRB, evaluating the extreme precipitation events using TRMM and CHIRPS datasets by comparing them with in-situ rain gauge network products is necessary. For example, the authors can compare three figures by showing the relationship between the accumulated frequency and daily precipitation classes from in-situ rain gauge observation, TRMM and CHIRPS. Through the comparison of such figures, the readers can understand not only basin-averaged properties in each precipitation product but also which products can be representative in terms of extreme events.    

 (4) The authors need to evaluate orographic precipitation effects because most in-situ rain gauges are located in lower flat areas. However, satellite-based precipitation can measure the precipitation in mountainous regions too. One of the reasons for the overestimation of satellite-based precipitation against in-situ data can be explained by the orographic effect on precipitation. In order to evaluate orographic effects on precipitation, the authors can divide the LMRB into individual tributary basins. The authors can then detect how the accuracy of satellite-based precipitation depends on the number of rain gauges and complex orography.   

(5) I cannot understand how such a substantial overestimation of satellite-based precipitation against in-situ observations can be useful for evaluating river runoff. To demonstrate the importance of TRMM and CHIRPS datasets, the authors need to use a river runoff model in LMRB, forced by in-situ observations, TRMM and CHIRPS precipitation datasets. If TRMM and CHIRPS precipitation does not produce reasonable river runoff in LMRB, the authors should indicate this.

Reviewer 3 Report

The paper "Evaluation of Satellite-Based Rainfall Estimates in the Lower Mekong River Basin" by Dandridge and co-workers presents a validation study of two satellite-based rainfall products (TMPA-3B42 and CHIRPS) over the lower Mekong River basin. For a 15-year perod, daily, monthly and yearly precipitation products are compared with rain values measured by a ground based raingauge network.

The paper is reasonably well written, with clear description of the work done, but, in my opinon, the information content is very low, and there are also mistakes that make me suggest its rejection.

First, the Authors consider CHIRPS and 3B42 as "satellite products", while it is clear that both of them are computed by using also raingauges data. This is mentioned for CHIRPS, but not for 2B42. In this paper, the product to be validated it depends on the reference field used to perform the validation, since raingauges data are used in the algorithms to calibrate, or adjust, the final product. In other words, what the Authors measure is not the capability of the product to detect precipitation, but how strong is the impact of the gauges calibration in the satellite-based algorithms. To overcome this point, the Authors should, at least, see how many and which raingauges are used in the calibration procedure in CHIRPS and in 3B42.

A further option could be to perform the validation only over interpolated gridpoints, excluding the points where raingauges are present, and compare the results with the numbers obtained in the present study. 

Other issues

line 67. Remote sensing does not "measure" precipitation, it measures radiance, that can be used to "estimate" precipitation.

line 68. As a matter of fact, radars are remote sensing instruments.

line 151 and in the rest of the manuscript. These are not satellite products, but, satellite-based products.

lines 162-165. To make this analysis meaningful, it has to be discussed which is the minimum detectable rainrate observed by the 3 products. How is set the threshold between dry and wet samples? A sample with monthly average 0.001 mm/month is wet or dry?

Figure 2. top right box: how are "invalid raingauge data" detected? which quality control of the gauge data is performed?

lines 181-182. This sentence dosn't tell anything,  please, explain the meaning of mean absolute error.

lines 186-187. How did you find this?

lines 204-209. Where are these number from? how are "accuracy" defined? how "falsely predicted rain" is computed?

The conclusions are very poor: the results are repeated without attempts to understand or explain the findings. Some example follows.

lines 296-298.This sentence is vague, and is not a result of the present paper.

lines 325-327. In this paper no original validation procedure is presented: it is the same validation methodology applied in tons of validation papers.  

lines 327-329. This statement is not proved in this study, where no "hydrologic cycle modeling" is done. In Mohammed et al. (2018) it was done.

lines 336-339. In the present paper issues as "topography, average rainfall, etc. in each sub-basin" are not addressed.

line 339. Ho the different ground resolution of the two products impacts on the results?

Finally, I disagree with the lase sentence of the paper. To be really useful to the hydrological community, this study should check how the "subtle differences" between products propagates in hydrological models.

Round 2

Reviewer 2 Report

Major comments

My comments relate to the manuscript by Chelsea Dandridge et al., entitled “Evaluation of Satellite-Based Rainfall Estimates in the Lower Mekong River Basin”. This manuscript compared daily rainfall estimates from TRMM Multi-Satellite Precipitation Analysis 3B42 v.7 and Climate Hazards Group InfraRed Precipitation with Station data (CHIRPS) against rain gauge observations in the Lower Mekong River Basin (LMRB) in Southeast Asia, from 2000 to 2014.

According to previous comments, the first version of the manuscript has been improved, however the quality is still insufficient for publication. The observation and interpolation methods (kriging method) are inadequate. I believe that the manuscript is unsuitable for publication in its current form, and requires major revisions and additional modifications prior to being considered for acceptance. My primary concerns are the following:

(1) Lines131–145: The objective of this study appears somewhat redundant and authors should clarify what the most important message from this paper is. In the current state, it is very difficult for readers to understand what this paper is focusing on. Please simplify and rewrite the objectives.

(2) Lines 215–217: Authors noted that the interpolation is simply a displayed spatial pattern of rainfall. If this is the case, please delete all interpolated rainfall figures from the manuscript as readers could misunderstand Figure 3. Please modify Figure 3, using the point values as in Figures 6 and 7. I disagree with showing interpolated results from an in-situ dataset, and satellite-based products should also be displayed along with in-situ datasets.   

(3) Line 249: The caption of Table 3 is not sufficient to understand what is listed in the table. Please add more information regarding the type of validation that is shown in Table 3.

(4) Figure 3: As described in point (2) above, all figures should be adjusted to the in-situ datasets, and should not show interpolated in-situ data such as satellite-based products.  

(5) Tables 5 and 6 should list the number of stations in each category range (for monthly rain accumulation and elevation, respectively), so that the reader may understand how the number of stations affects the statistical values.

(6) Figures 6 and 7: These two figures should use the same colors in accordance with the same R-value range, so that the reader may understand how spatial correlation was distributed by the CHIRPS and TRMM precipitation estimates.

Reviewer 3 Report

Second review

I will comment on the Authors’ replies and then I will introduce other issues that makes the present manuscript not suitable for publication on Remote Sensing.

The paper "Evaluation of Satellite-Based Rainfall Estimates in the Lower Mekong River Basin" by Dandridge and co-workers presents a validation study of two satellite-based rainfall products (TMPA-3B42 and CHIRPS) over the lower Mekong River basin. For a 15-year perod, daily, monthly and yearly precipitation products are compared with rain values measured by a ground based raingauge network. The paper is reasonably well written, with clear description of the work done, but, in my opinon, the information content is very low, and there are also mistakes that make me suggest its rejection.

Response: Thank you for your time and consideration of this manuscript. We have added additional analyses and results in order to expand the information content of this manuscript. We have also thoroughly proofread and corrected mistakes throughout the paper.

Comment: My overall feeling is that the information content of the paper is not changed and several flaws are still present in the manuscript. Many previous comments are not fully addressed in the reviewed manuscript.

First, the Authors consider CHIRPS and 3B42 as "satellite products", while it is clear that both of them are computed by using also raingauges data. This is mentioned for CHIRPS, but not for 2B42. In this paper, the product to be validated it depends on the reference field used to perform the validation, since raingauges data are used in the algorithms to calibrate, or adjust, the final product. In other words, what the Authors measure is not the capability of the product to detect precipitation, but how strong is the impact of the gauges calibration in the satellite-based algorithms. To overcome this point, the Authors should, at least, see how many and which raingauges are used in the calibration procedure in CHIRPS and in 3B42.

Response: Searching through the sources of climate data used in CHIRPS (from https://iridl.ldeo.columbia.edu/SOURCES/?Set-Language=en), two sets of in situ data are used, Agromet Group of the Food and Agriculture Organization of the United Nations (FAO) and Global Historical Climate Network (GHCN). These two data sets are long-term averages and were used to create the CHPclim. The stations’ historical data was used in the calibration for the CHIRPS method not data from the study period of this work. For example, zero stations provided data for the GHCN in Cambodia, one station in Laos, 10 stations in Thailand, and 4 in Vietnam from 2000 to 2014.

Comment: I accept this reply for CHIRPS, but this should be clearly written in the manuscript. No mention of the use of raingauges in the 3B42 algorithm is present. Moreover, CHIRPS makes use of 3B42 product to evaluate some cloud properties, so the two products considered here share much information, and this should be clearly told to the reader.

A further option could be to perform the validation only over interpolated gridpoints, excluding the points where raingauges are present, and compare the results with the numbers obtained in the present study.

Response: This is outside the scope of this study.

Comment: ok, it was just a suggestion to increase the very low information content and novelty of the paper.

Finally, I disagree with the lase sentence of the paper. To be really useful to the hydrological community, this study should check how the "subtle differences" between products propagates in hydrological models.

Response: Thank you for your comment. The last line has been changed, “This study is a unique look at comparing TRMM and CHIRPS datasets with in-situ observations in the LMRB and provides useful insight to their performance that should be considered when applying them.”

Comment: This is a vague and useless sentence: how the results of this paper can be used? Did you provide accuracy or reliability quantitative information? How your results would be really useful for the community? To say that the “accuracy” of the products, whatever it is, is 55.8% or 49.3%, would prevent anyone to use this product for quantitative application.

Other issues

lines 162-165. To make this analysis meaningful, it has to be discussed which is the minimum detectable rainrate observed by the 3 products. How is set the threshold between dry and wet samples? A sample with monthly average 0.001 mm/month is wet or dry?

Response: The following sentence has been changed to clarify the methodology, “This was done by determining how many times the satellite-based estimate and the gauge measurement in a particular pixel were both wet (not zero) or dry (zero) and if one was wet and the other was dry (Table 1). “

Comment: The point was not properly addressed. My question was: which is the threshold between zero-rain and non-zero-rain? Again: A sample with daily amount of 0.03 mm, as detected by TMPA (Table 2) can be safely defined as “wet” sample? This should be addressed in the data description section, where the minimum detectable quantity for the different dataset should be mentioned. Especially for the raingauge, it is important to know which type of instrument are used, which quality control is performed (see next point).

Figure 2. top right box: how are "invalid raingauge data" detected? which quality control of the gauge data is performed?

Response: We added the following lines: “The original in-situ data contained values that did not represent valid measurements. These measurements with values of -9999 were changed to no data during the pre-processing of the ground measurements.”

Comment: The Authors bypassed the real question: how pre-processing is performed? How did you recognize invalid measurements? Who and how put -9999 for no-data?

lines 181-182. This sentence doesn't tell anything, please, explain the meaning of mean absolute error.

Response: We changed the sentence to the following: “The Mean Absolute Error (MAE) gives the average of the absolute errors and measures the difference between two continuous variables.”

Comment: ok, but all error indicators should be defined here (e.g. bias, or mean error). Moreover, the binary indicators introduced in table 2 are not the ones commonly used for this analysis (i.e. POD, FAR, ETS…), see Nurmi, 2003: Recommendations on the verification of local weather forecasts. ECMWF Technical Memoranda. Technical Memorandum No. 430. If the Authors want to introduce new indicators, should well define their meaning in the text. Finally, the number D should always be zero, according to lines 185-187, right? So that should be removed from the table.

lines 186-187. How did you find this?

Response: We changed the sentence to the following: “When compared to other interpolation methods such as IDW and spline, we found that the kriging method resulted in the most accurate and most representative maps of the overall precipitation pattern for the data used in this study.”

Comment: the sentence is changed, but the question remains: how the “accuracy” and the “representativeness” of the kriging with respect to others? I was simpler to stop the text at line 209 of the new manuscript.

lines 204-209. Where are these number from? how are "accuracy" defined? how "falsely predicted rain" is computed?

Response: We have inserted the following table: Table 1. Validation Methodology

Comment: the “accuracy” is not defined (see lines 366-367): what is 49.3%? I can’t find this number in any Table in the paper.

The conclusions are very poor: the results are repeated without attempts to understand or explain the findings. Some example follows.

lines 296-298.This sentence is vague, and is not a result of the present paper.

Response: Thank you for this comment. This sentence has been changed, “Similarly, the monthly comparisons between in-situ and both satellite-based data sets showed higher correlations than the yearly comparisons when analyzed over the fifteen-year study period.”

Comment: you simply replaced a comment on an interesting point with the n-th vague sentence. Why not to understand the role of orography, data availability and precipitation amount?

lines 325-327. In this paper no original validation procedure is presented: it is the same validation methodology applied in tons of validation papers.

Response: The following lines have been changed to include the methodology and results from similar studies, “The hydrologic significance of TRMM and CHIRPS in the LMRB can be assessed from the results of this study and analogous validation studies. In addition, a similar methodology to the one described here can be applied to GPM IMERG data to further assess performance of the satellite-based precipitation products in the region.”

Comment: The point here was in this paper there is not new methodology.

lines 327-329. This statement is not proved in this study, where no "hydrologic cycle modeling" is done. In Mohammed et al. (2018) it was done.

Response: Thank you for your comment. The lines regarding the implications in “hydrologic cycle modeling" have been removed as they were not proven in this study.

Comment: ok, very low impact of the results of this paper for hydrology.

lines 336-339. In the present paper issues as "topography, average rainfall, etc. in each sub-basin" are not addressed.

Response: These lines have been changed, “For future validation studies of satellite based estimates, this methodology can be applied to newer, higher resolution products like GPM IMERG to look at the progression and advancement in satellite-based estimation.”

Comment: again, this is a vague sentence referring to future work.

line 339. How the different ground resolution of the two products impacts on the results?

Response: Thank you for pointing this out. We have added the following to the manuscript -

The spatial resolution does play a role in the validation of precipitation products. CHIRPS has a spatial resolution of 0.05o and TRMM is at 0.25o. Generally, higher spatial resolution translates to higher accuracy, but this is dependent on the method used to generate this product. What we imply is an inferior method used to generate a high spatial resolution product that may have a lower accuracy than a superior method used to generate a lower spatial resolution product. In this study we find that the accuracy of TRMM and CHIRPS are very close and the higher spatial resolution of CHIRPS does not provide an advantage in the accuracy when compared to rain gages.

Comment: The spatial resolution affects both the accuracy AND the results of the validations. The fact that “generally higher spatial resolution translates to higher accuracy” is questionable and not proved (any reference?)

This was about my first review. Below I list more comments on the new version of the manuscript.

15.Introduction. The first part (lines 42-69) is too long and addresses topics not interesting for this paper (i.e. crops, fishing, hydropower) and should be shortened.

16.line 86 and all over the text. Satellite precipitation are not data, but products.

17. lines 103-105. This is highly questionable: 1) the proposed methodology does not show anything new, 2) IMERG is a totally different product in term of algorithms, data used and resolution. Furtherly, TRMM is the name of the satellite mission, the products can be referred as TMPA (TRMM Multi-satellite Precipitation Analysis).

line 132. How TMPA and CHIRPS could “predict” precipitation? line 139 IMERG is available since March 2014. lines 157-168. The two satellite-based products have to be described with more details, highlighting the use of common ancillary data. Where are the results of these indicators (correct detection, incorrect detection,…)? in Table 3 there are 3.3% of samples where no rain is found by both TMPA and gauges, but elsewhere it is said that the samples with no-rain are not considered. From where does this 3.3% come?